# Histone variant H2A.Z modulates nucleosome dynamics to promote DNA accessibility

Shuxiang Li[1], Tiejun Wei [1] & Anna R. Panchenko [1,2,3,4] ✉

Nucleosomes, containing histone variants H2A.Z, are important for gene transcription initiation and termination, chromosome segregation and DNA double-strand break repair, among other functions. However, the underlying mechanisms of how H2A.Z influences nucleosome stability, dynamics and DNA accessibility are not well understood, as experimental and computational evidence remains inconclusive. Our modeling efforts of human nucleosome stability and dynamics, along with comparisons with experimental data show that the incorporation of H2A.Z results in a substantial decrease of the energy barrier for DNA unwrapping. This leads to the spontaneous DNA unwrapping of about forty base pairs from both ends, nucleosome gapping and increased histone plasticity, which otherwise is not observed for canonical nucleosomes. We demonstrate that both N- and C-terminal tails of H2A.Z play major roles in these events, whereas the H3.3 variant exerts a negligible impact in modulating the DNA end unwrapping. In summary, our results indicate that H2A.Z deposition makes nucleosomes more mobile and DNA more accessible to transcriptional machinery and other chromatin components.

In eukaryotes, genomic DNA is packaged into chromatin using the building blocks called nucleosomes, which also serve as hubs in epigenetic signaling pathways. The nucleosome core particle consists of approximately 147 base pairs of DNA wrapped around an octamer of histone proteins[1]. Histones have been historically divided into 'canonical' histones that are expressed only during the S-phase of the cell cycle and 'variants' that are constitutively expressed during the cell cycle, among other distinguishing features[2]. Substitution of canonical histones with histone variants is one of many fundamental ways to dynamically regulate the function of chromatin.

Histone variant H2A.Z belongs to the histone H2A family and is broadly distributed in eukaryotes. H2A.Z family is evolutionarily conserved, but human H2A.Z shares only about 60% identity with the canonical H2A[3]. H2A.Z predominantly accumulates in the upstream and downstream of the transcription start sites (TSS) and is enriched in +1 nucleosome in gene bodies, as well as in enhancer elements of active genes or transcriptionally poised genes[4–8]. Given the critical location of H2A.Z-containing nucleosomes in genome and chromatin, one might think that these nucleosomes have major roles in transcription. Indeed, the H2A.Z variant is well known for its functions in transcription initiation and termination[9–11], but there is no unifying view on how H2A.Z incorporation into nucleosomes can mediate the transcription-related processes. On one hand, it has been previously reported that H2A.Z depletion leads to RNA polymerase pausing by the nucleosomal barrier at the transcription start site[12,13]. On another hand, the incorporation of H2A.Z may decrease but widen the barrier to transcription[14] and slow the rate of RNAPII pause release[15]. In addition, many experimental studies reveal that H2A.Z can accumulate in facultative heterochromatin regions and areas of silenced genes[16,17] and could be involved in heterochromatin boundaries[18] and chromosome segregation[19], facilitate DNA double-strand break repair[20], and cell cycle progression[21]. The importance of H2A.Z for clinical research is evident from the observations that H2A.Z overexpression is linked to poor prognosis in prostate, breast, bladder, liver, and lung cancers[22].

The functional implications of H2A.Z can be revealed by studying its physico-chemical properties, stability, and dynamics. However, the

[1]Department of Pathology and Molecular Medicine, Queen's University, Kingston, ON, Canada. [2]Department of Biology and Molecular Sciences, Queen's University, Kingston, ON, Canada. [3]School of Computing, Queen's University, Kingston, ON, Canada. [4]Ontario Institute of Cancer Research, Toronto, Canada. ✉e-mail: anna.panchenko@queensu.ca

findings of these studies are also filled with conflicting reports. Various experimental works have indicated that the H2A.Z-containing nucleosomes can be less or more stable than the canonical ones[23–25], whereas X-ray crystal structures of canonical and H2A.Z nucleosomes show very similar overall conformations except for the L1 loop region[26]. Furthermore, to make the story more complex, it was reported that the stability and dynamics of H2A.Z nucleosomes depended on the organism and nucleosomal context, such as the presence of other histone variants[27,28] or histone post-translational modifications (PTMs)[29]. For example, previous studies have shown that nucleosomes containing double variants of H3.3 and H2A.Z were enriched at transcription start site (TSS) in human cells and facilitated the access of transcription factors to the packaged DNA[30].

Two major questions remain unanswered in this respect. First, does the deposition of H2A.Z variant influence the nucleosome stability, dynamics and DNA unwrapping? Second, how do these mechanisms couple with the role of H2A.Z in gene expression and other biological functions? To address these questions, we have performed in silico studies by modeling the dynamics and energetics of human heterotypic and homotypic nucleosome systems containing H2A.Z variants (with and without H3.3) and canonical histones. Our results, corroborated in multiple seven-microsecond long molecular dynamics simulation runs, show that incorporation of the H2A.Z variant enhances DNA and histone dynamics. Namely, H2A.Z C-terminal tail and histone core plasticity mediate the pronounced DNA unwrapping, whereas the N-terminal tail accounts for the increased nucleosome gapping.

## Results

### H2A.Z deposition facilitates the DNA unwrapping

To investigate the role of H2A.Z in nucleosome dynamics, we first constructed two homotypic nucleosome systems, containing canonical H2A (NUC$_{H2A/H2A}$, Fig. 1a) and H2A.Z variant (NUC$_{H2A.Z/H2A.Z}$), and carried out three independent 7 μs long all-atom molecular dynamics (MD) simulations for each system (see Supplementary Fig. 1 for the description of initial structural models and Supplementary Table 1 for simulation setup details). We used 187 bp of the native DNA sequence of *TP53* gene, 147 bp of nucleosomal sequence and two linker DNA segments of 20 bp long each. First, as shown in Fig. 1b, c, in both canonical and variant systems one DNA end unwraps, while another end stays bound to a histone octamer. Previous single-molecule FRET experiments and MD simulations demonstrated that DNA unwrapping indeed occurred asymmetrically and DNA unwrapping at one side stabilized the DNA end on the other side[31–33]. Moreover, we observe the difference in terms of the degree of unwrapping for both DNA ends that can be related to the asymmetry of our nucleosomal DNA sequence with respect to the dyad position. Second, we find that the overall DNA unwrapping from both ends is significantly enhanced in H2A.Z (up to 45 bps unwrapped in total from both ends) compared to canonical H2A nucleosomes (up to 22 bps from both ends) (Fig. 1d). Previous microsecond-long simulations and quantitative modeling showed that canonical nucleosomes exhibited spontaneous DNA unwrapping for systems without histone tails[32–34]. The roles of histone tails in DNA unwrapping will be discussed in the next sections.

Although the effects of other histone variants are not the focus of our study, we performed six additional MD simulations of two nucleosome systems with the H3.3 variant as it is often deposited in H2A.Z nucleosomes (Supplementary Fig. 1 and Supplementary Table 1). Our results reveal that both homotypic variant nucleosomes with two copies of H2A.Z (NUC$_{H2A.Z/H2A.Z}$ or NUC$_{H2A.Z/H2A.Z+H3.3/H3.3}$, Fig. 1c and Supplementary Fig. 4a) show similar amplitudes of DNA unwrapping indicating a relatively small effect of H3.3 variant on nucleosome dynamics. Moreover, homotypic H2A.Z nucleosomes exhibit a larger magnitude of DNA unwrapping as compared to heterotypic H2A.Z nucleosomes with one copy of HA2.Z, no matter how many copies of H3.3 are present (NUC$_{H2A/H2A.Z+H3/H3.3}$,

Supplementary Fig. 4b). Interestingly, although the distributions of the number of unwrapped base pairs are very similar for NUC$_{H2A.Z/H2A.Z}$ or NUC$_{H2A.Z/H2A.Z+H3.3/H3.3}$ systems (Fig. 1d and Supplementary Fig. 4c), the deposition of H3.3 causes a somewhat more pronounced unwrapping of the entry side of DNA which is less mobile in NUC$_{H2A.Z/H2A.Z}$ systems (Fig. 1c). Overall, we can conclude that H2A.Z has the strongest effect in modulating the DNA unwrapping, and the H3.3 variant may play a less prominent role. Our result is in agreement with previous X-ray structure studies showing that the incorporation of histone H3.3 did not affect the structure of the H2A.Z-containing nucleosome[25]. However, it was also reported that the effect of H2A.Z on nucleosome stability was strongly dependent on the presence of H3.3[27].

### Energy barrier for DNA unwrapping is reduced in H2A.Z systems

We further computed the free-energy profile as a function of the number of unwrapped base pairs to estimate the energy barriers in the process of DNA unwrapping. Our results show that in the canonical H2A nucleosome, an average energy of ~2 and ~6 kcal/mol is needed to unwrap ~5 and ~17 base pairs of the nucleosomal DNA (the maximum degree of unwrapping from one DNA end observed on our time scale) with respect to the initial structure (Fig. 1e). This is in good agreement with the unwrapping energy of ~17 bp of DNA reported earlier[35,36]. Other experimental forced disassembly studies on single-molecule unzipping of nucleosomal DNA pointed to the free energy of dissociation of the outside 76 bp being about 12 kcal/mol[37] and also detected the first energy barrier around the position 17 bp in canonical nucleosomes[14].

Next, we show that the incorporation of variant H2A.Z significantly reduces the energy barrier by several kcal/mol compared to the canonical nucleosome with a wider range of unwrapping modes (Fig. 1e), consistent with the previous experimental estimates[14]. Figure 1f, g presents the two-dimensional free-energy landscape as a function of the DNA radius of gyration ($R_g$) and the total number of DNA-histone contacts. Canonical nucleosome exhibits a smaller range of $R_g$ compared to the variant nucleosome which is coupled with a larger number of DNA-histone contacts. The landscapes clearly illustrate that the energy cost of opening the two DNA ends in NUC$_{H2A.Z/H2A.Z}$ is considerably smaller than that in NUC$_{H2A/H2A}$. To verify these findings, we performed the MM/GBSA calculations showing that the overall absolute value of histone-DNA binding energy of the NUC$_{H2A.Z/H2A.Z}$ system is lower than that of the NUC$_{H2A/H2A}$ system (Supplementary Table 4), indicating that the incorporation of H2A.Z disfavors the DNA-histone binding, which is consistent with their unwrapping characteristics described above. The reduced stability of H2A.Z-containing nucleosomes was observed previously[38].

### H2A.Z C-terminal tail modulates the DNA unwrapping

In the first section, we showed that DNA ends of H2A.Z-containing nucleosomes are more mobile and are easier to unwrap from the histone octamer when compared to nucleosomes containing canonical H2A. It has been reported earlier that H3 N-terminal and H2A C-terminal tails are important in DNA breathing (refers to a less pronounced DNA unwrapping of ~5–10 bp) in canonical nucleosomes[32,33,39–43]. Here, we show that the degree of DNA unwrapping is anti-correlated with the number of contacts between the H2A.Z C-terminal tail and the outer DNA region (Fig. 2a, see Supplementary Fig. 5 for definitions of DNA regions). Such a pattern is observed in all three independent simulation runs (Supplementary Fig. 6).

To further test the causation of this effect, we conducted simulations of systems with the swapped H2A and H2A.Z C-terminal tails (Supplementary Table 2 and Supplementary Fig. 8). Compared with canonical nucleosomes (NUC$_{H2A/H2A}$), the nucleosome containing H2A with H2A.Z C-terminal tail (H2A$_{H2AZtail}$) exhibits a larger magnitude of DNA unwrapping on the same simulation time scale (2 μs, Fig. 2f, Supplementary Fig. 9). Moreover, the H2A.Z nucleosome with H2A

C-terminal tails (H2AZ$_{H2Atail}$) shows a considerably lower magnitude of DNA unwrapping as compared to NUC$_{H2A.Z/H2A.Z}$ (Supplementary Fig. 9). Based on these observations, we conclude that the H2A/H2A.Z C-terminal tail is a very important factor that determines the DNA unwrapping from the histone octamer.

Next, we speculate that the differences in DNA unwrapping between the canonical and H2A.Z nucleosomes might be due to the distinct features in the human H2A.Z C-terminal tail. Indeed, in human, in comparison with the H2A C-terminal tail (residues 121–130), the H2A.Z C-terminal tail (residues 123–128) is shorter and lacks several positively charged residues making it less prone to interact with DNA (Fig. 3a). The analysis of the conformational ensemble of the tails from three simulation runs shows that the average distance between the DNA end and C-terminal tail in

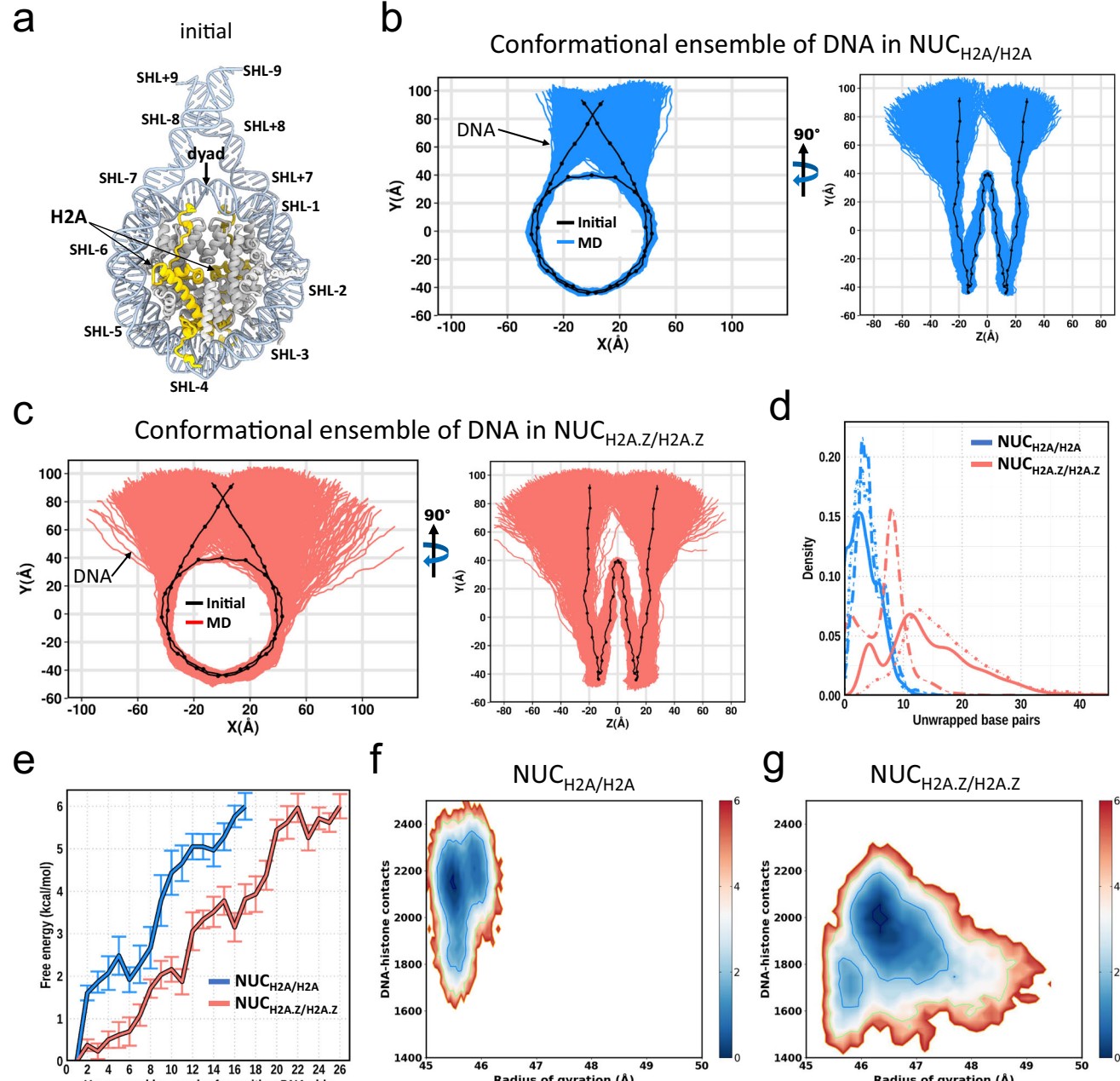

**Fig. 1 | H2A.Z facilitates nucleosomal DNA unwrapping. a** Cartoon representation of the initial nucleosome structure, histone tails of H3, H4, and H2B are not shown. Superhelical locations (SHLs) where major grooves face the histone surface are labeled. Histone H2A is shown in yellow and other histones are shown in gray. **b** 2D projection of DNA conformations for NUC$_{H2A/H2A}$ (blue) from the combined ensemble of three independent simulation runs, 7 μs each (see Supplementary Fig. 3 for results for each simulation run). The black dotted line represents the initial conformaion. **c** Same as **b** but for NUC$_{H2A.Z/H2A.Z}$ (red). **d** Distributions of a total number of unwrapped base pairs of NUC$_{H2A/H2A}$ (blue) and NUC$_{H2A.Z/H2A.Z}$ (red) nucleosomes for three independent simulation runs (solid, dotted and dashed lines). The total number of unwrapped base pairs is a sum of the number of unwrapped base pairs from both DNA sides. Unwrapped base pairs are counted from the beginning of nucleosomal DNA (see Supplementary Fig. 5 for definitions of DNA regions). **e** Free-energy profile as a function of the number of unwrapped base pairs for NUC$_{H2A/H2A}$ (blue) and NUC$_{H2A.Z/H2A.Z}$ (red) nucleosomes. The error bars represent standard errors of the free energy for two DNA sides from three independent simulation runs ($n = 6$). The $x$-axis goes to the maximum number of DNA base pairs unwrapped from one end. **f** Two-dimensional projection of the free-energy surface as a function of the DNA radius of gyration and the total number of DNA-histone contacts for NUC$_{H2A/H2A}$ nucleosome combined from three independent simulation runs. Energy bar values are shown in kcal/mol. **g** Same as **f** but for NUC$_{H2A.Z/H2A.Z}$ nucleosome.

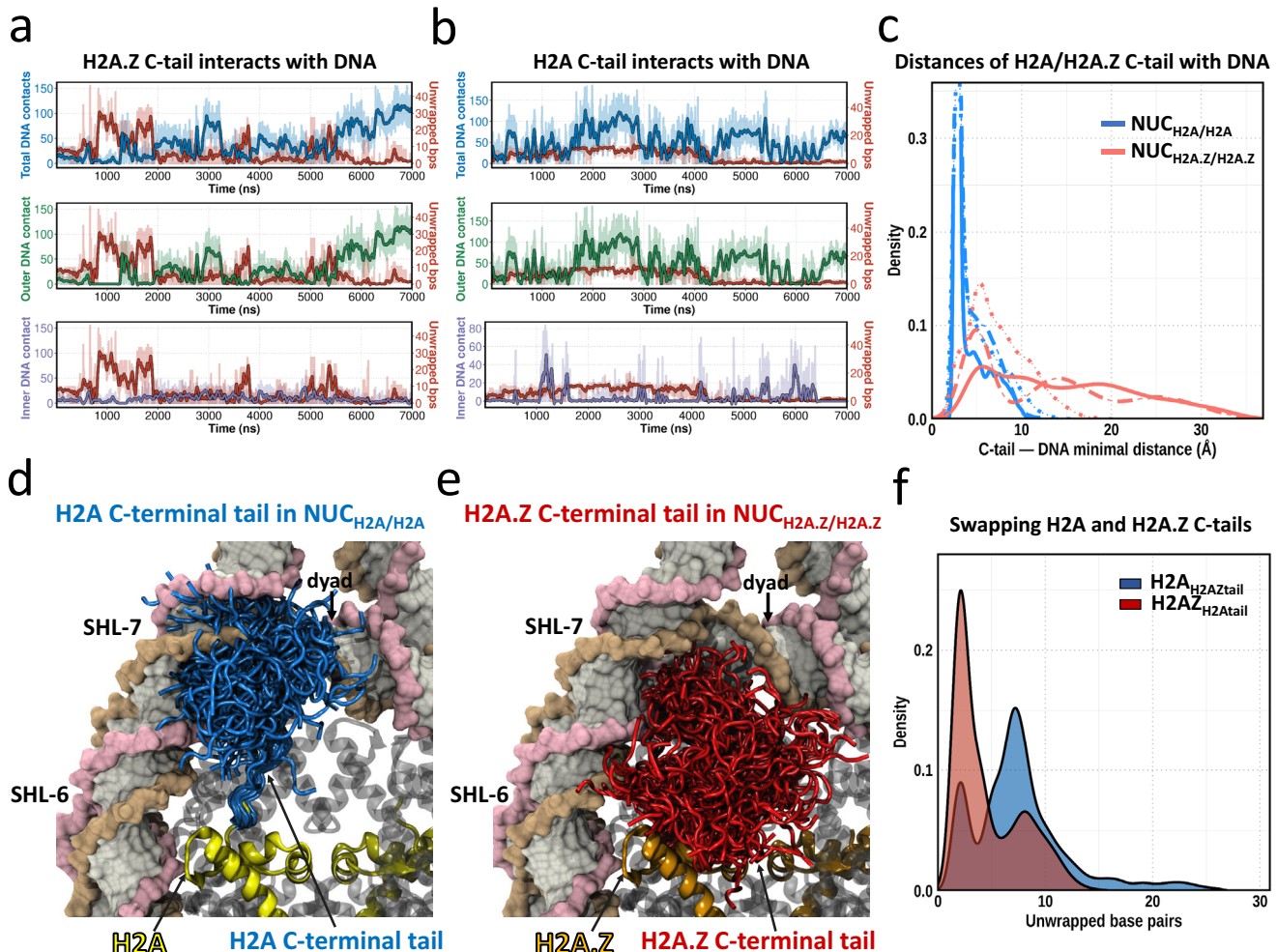

**Fig. 2 | H2A.Z C-terminal tail mediates the DNA unwrapping. a** A representative simulation run shows the coupling between the number of unwrapped base pairs from both DNA ends (red) and the number of H2A.Z C-terminal tail−DNA contacts in $NUC_{H2A.Z/H2A.Z}$ nucleosome. The number of contacts of H2A.Z C-terminal tail with total DNA, outer DNA and inner DNA regions are shown in blue, green and purple, respectively. See Supplementary Fig. 6 for other simulation runs. Lines with faded colors correspond to raw data and are smoothed with a Savitzky-Golay filter using a ten ns window and first-degree polynomial (dark color lines). **b** Same as **a** but for $NUC_{H2A/H2A}$ nucleosome. See Supplementary Fig. 7 for other simulation runs. **c** Distribution of distances between H2A (blue) or H2A.Z (red) C-terminal tail and DNA segment (base pair from ±41 to ±75) for three independent simulation runs (solid, dotted and dashed lines). **d** Structural conformations of H2A C-terminal tail (blue color, residues 121–130) in the canonical nucleosome ($NUC_{H2A/H2A}$) combined from three simulation runs. All frames are taken every 50 ns. **e** Structural conformations of H2A.Z C-terminal tail (red color, residues 123–128) in H2A.Z nucleosome ($NUC_{H2A.Z/H2A.Z}$) combined from three simulation runs. **f** Distributions of a total number of unwrapped base pairs of $H2A_{H2AZtail}$ (blue) and $H2AZ_{H2Atail}$ (red) on the time scale of two microseconds. See Supplementary Fig. 9 for DNA conformational ensembles for $H2AZ_{H2Atail}$ and $H2A_{H2AZtail}$ systems.

$NUC_{H2A/H2A}$ is much smaller than that in $NUC_{H2A.Z/H2A.Z}$ (4.1 ± 0.5 Å versus 12.4 ± 3.9 Å, Fig. 2c). This indicates that extensive interactions are formed between the H2A C-terminal tail and DNA with contacts predominantly distributed in DNA regions at superhelical locations SHL ± 6 and SHL ± 7 (Fig. 2d, see Methods for definition of superhelical locations). As a result of the loss of interactions with DNA, the H2A.Z C-terminal tail almost entirely protrudes from the nucleosome disc into the solvent with only a few tail conformations interacting with the DNA end (Fig. 2e). These results suggest that the absence of positively charged residues at the end of the tail in H2A.Z dramatically weakens the association between the H2A.Z C-terminal tail and DNA within the same nucleosome, leading to DNA unwrapping. Consistent with this, a recent single-particle cryo-EM study suggested that the H2A.Z C-terminal tail was more flexible than the canonical H2A[44]. Interestingly, we find that the protruded C-terminal tail of H2A.Z is long enough to span the nucleosome-nucleosome distance in the nucleosome arrays[45] (Supplementary Fig. 10).

## H2A.Z deposition increases the plasticity of histone octamer

The change in structural dynamics of the histone core regions upon H2A.Z deposition may be also associated with the DNA unwrapping. Figure 3b, c shows the structural variation of different regions of the core H2A/H2A.Z and H2B (excluding histone tails) with respect to the average structure, measured as root mean square fluctuations (RMSF). In both H2A.Z/H2A and H2B histones, the L1 and L2 loop regions show higher fluctuations than α-helices. The αN helix and αC helix of H2A.Z/H2A that are located near the N- and C-termini also exhibit high RMSF values. These observations are in good agreement with the H2A-H2B dimer plasticity in the context of nucleosomes observed in a recent MD study[46]. Although there is no obvious difference in RMSF for histone H3 and H4 core regions between $NUC_{H2A/H2A}$ and $NUC_{H2A.Z/H2A.Z}$ nucleosomes (Supplementary Fig. 11), a noticeable increase in structural fluctuations is observed for H2A.Z core regions in $NUC_{H2A.Z/H2A.Z}$ compared to the canonical

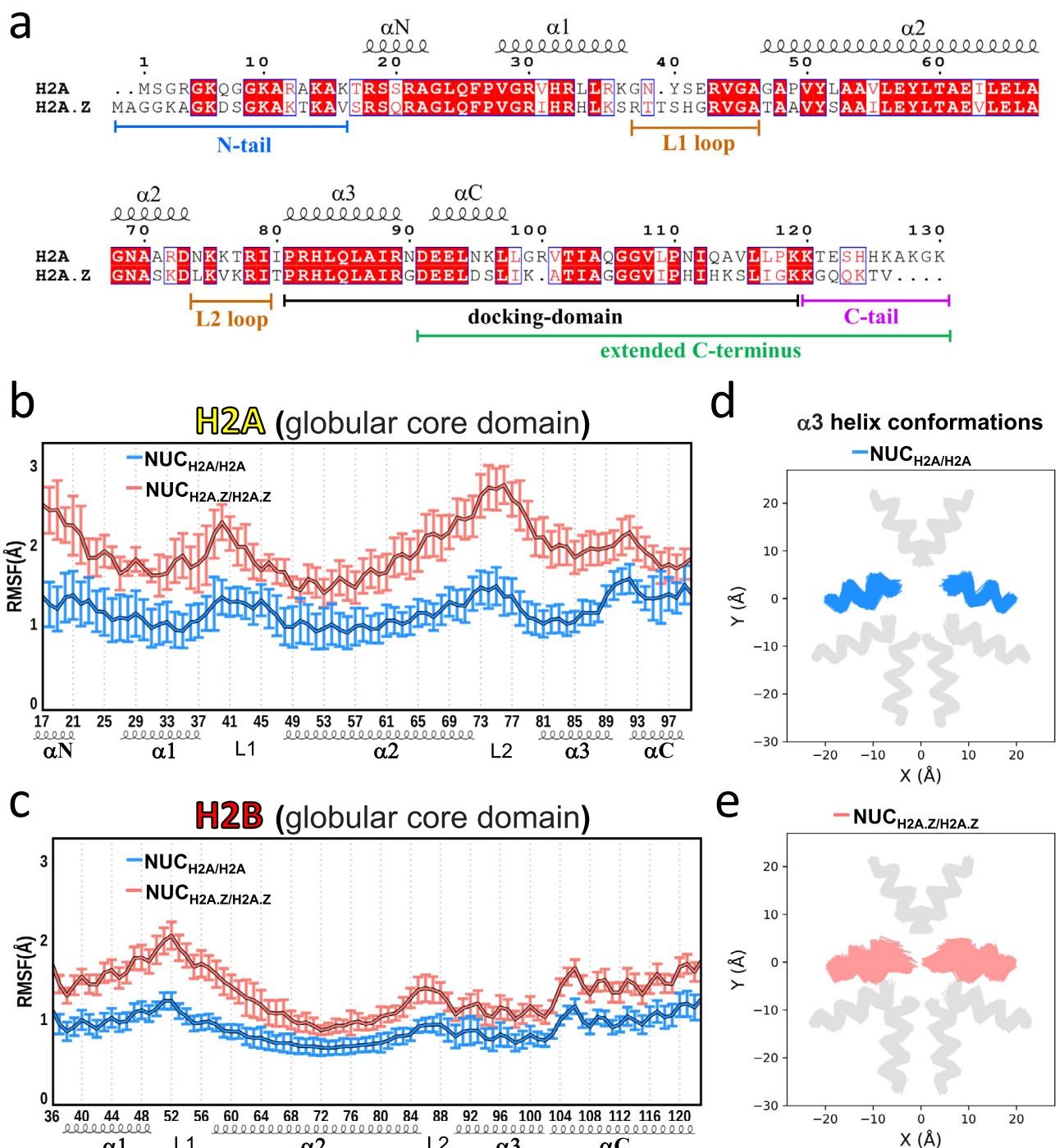

**Fig. 3 | H2A.Z incorporation increases the plasticity of histone octamer.**
**a** Sequence comparison between human H2A and H2A.Z.1. Identical residues are rendered as white characters on a red background. Similar residues are rendered as red characters and boxed. **b** The values of root-mean square fluctuations (RMSF) for Cα atoms of histone H2A globular core domain in NUC$_{H2A/H2A}$ nucleosome (blue) and H2A.Z in NUC$_{H2A.Z/H2A.Z}$ nucleosome (red). The error bars represent standard errors of RMSFs (n = 6) calculated from three independent simulation runs for two copies of H2A (NUC$_{H2A/H2A}$) or H2A.Z (NUC$_{H2A.Z/H2A.Z}$). **c** Same as **b** but for histone H2B globular core domain. See Supplementary Fig. 11 for the results of histones H3 and H4. **d** Conformational ensemble of H2A α3 helix in NUC$_{H2A/H2A}$ nucleosome (blue) from three independent simulation runs. Other types of histones are shown in gray. See Supplementary Fig. 12 for results of other α helices. **e** Same as **d** but for H2A.Z α3 helix in NUC$_{H2A.Z/H2A.Z}$ nucleosome (red).

H2A in NUC$_{H2A/H2A}$ (Fig. 3b). Interestingly, the structural fluctuation of histone H2B is also enhanced in NUC$_{H2A.Z/H2A.Z}$ nucleosomes (Fig. 3c) possibly because of the dimer formation between H2A.Z and H2B and long-range distance effects. Moreover, swapping the H2A and H2A.Z C-terminal tails doesn't impact the dynamics of the core regions of H2A/H2A.Z and H2B (Supplementary Fig. 14 and Fig. 3). This result suggests that the overall

octamer plasticity is enhanced in the H2A.Z nucleosome, which in turn can be associated with the nucleosomal and linker DNA dynamics. To further test this hypothesis, we performed an additional simulation with the restrained histone core Cα-atoms (H2AZ$_{restraint}$, see Methods) to estimate the effects of the histone core restraints on DNA dynamics. As shown in Supplementary Fig. 13, restraining the histone core plasticity results in a lower

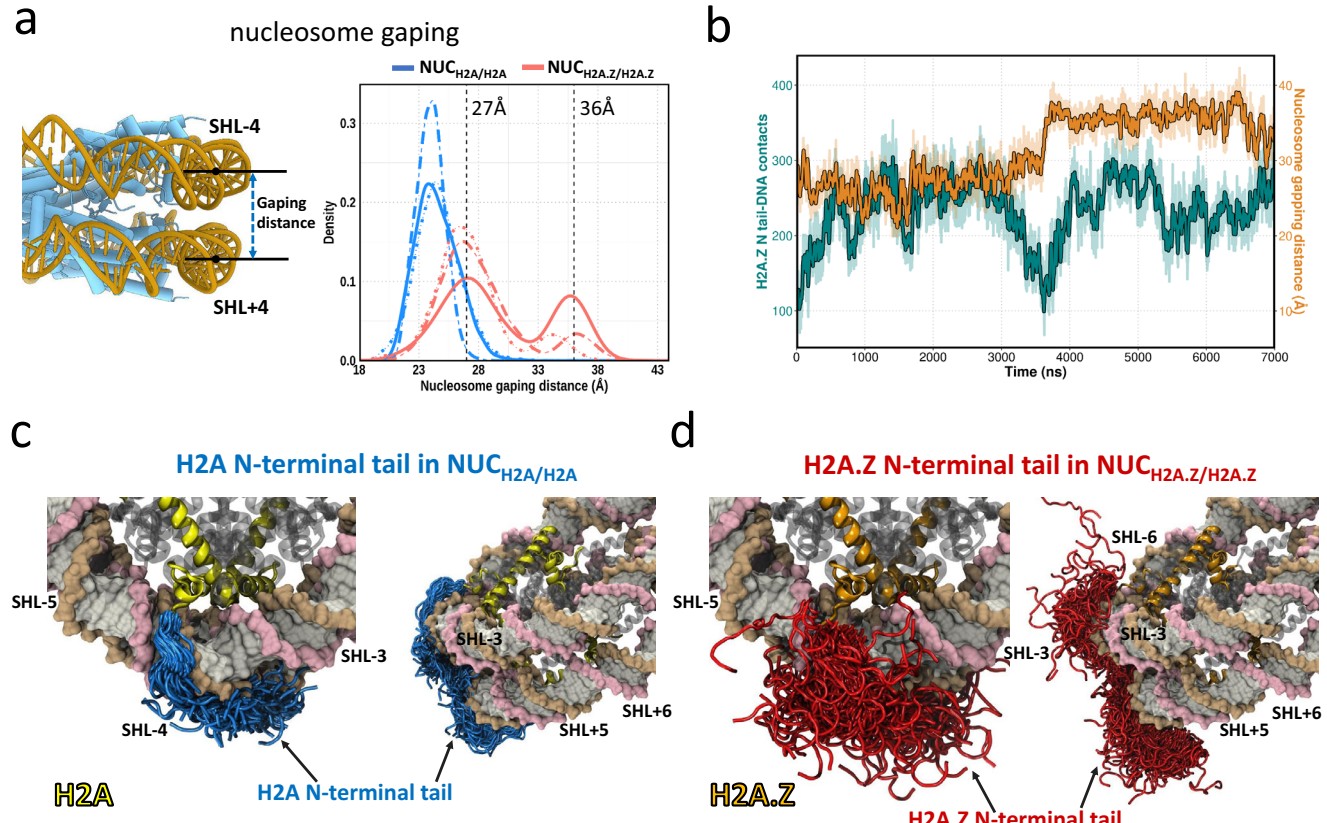

**Fig. 4 | H2A.Z N-terminal tail modulates nucleosome gapping. a** Distributions of nucleosome gaping distances in $NUC_{H2A/H2A}$ (blue) and $NUC_{H2A.Z/H2A.Z}$ (red) systems from three independent simulation runs (solid, dotted and dashed lines). Vertical dashed lines indicate the gaping distances at SHL ± 2 in the presence of SWR1 during the DNA translocation[50]. **b** A representative simulation run shows the time evolution of the H2A.Z N-terminal tail−DNA contacts (green) and nucleosome gapping distances (orange) in $NUC_{H2A.Z/H2A.Z}$. See Supplementary Figs. 15, 16 for time evolution for all three simulation runs for $NUC_{H2A.Z/H2A.Z}$ and $NUC_{H2A/H2A}$. **c** Structural conformations of H2A N-terminal tail (blue color, residues 1-16) in the canonical nucleosome ($NUC_{H2A/H2A}$) from three simulation runs. All frames are taken every 50 ns. **d** Structural conformations of H2A.Z N-terminal tail (red color, residues 1-18) in H2A.Z nucleosome ($NUC_{H2A.Z/H2A.Z}$) from three simulation runs.

magnitude of DNA unwrapping compared to the unrestrained $NUC_{H2A.Z/H2A.Z}$ on the same simulation time scale (2 µs). According to several recent studies, restricting the plasticity of histone octamers suppressed DNA fluctuations and led to a lower magnitude of DNA unwrapping and breathing[33,47,48].

**Gaping motions are enhanced in H2A.Z nucleosomes**
In addition to the enhanced DNA unwrapping induced by the incorporation of histone variant H2A.Z, another striking alteration is the noticeable spontaneous nucleosome gaping. Nucleosome gaping motions are distinct from DNA unwrapping. During the nucleosome gaping transitions, two half-turns (SHL+ and SHL−) of nucleosomal DNA open apart along the axis which is perpendicular to the nucleosome plane[49]. To quantitatively characterize the nucleosome gapping, we measured the distance between the two DNA segments in SHL-4 and SHL + 4 regions (Fig. 4a). In the canonical nucleosome, no significant gaping event is observed and the distribution of gaping distances peaks at ~24 Å for all three 7 µs simulation runs (Fig. 4a). However, the distance distributions between two DNA gyres are shifted to larger values in the H2A.Z nucleosomes with two peaks at ~27 Å and 36 Å, suggesting that the nucleosome gaping motions are substantially enhanced by the H2A.Z variant. Moreover, we find that restriction of histone core plasticity has a negligible impact on nucleosome gapping (Supplementary Fig. 17). Previous studies showed that during the DNA translocation of chromatin remodeler SWR1, the gaping distances measured between SHL ± 2 widened roughly from 27 Å to 36 Å in the presence of SWR1[50]. Although these studies were

conducted on canonical nucleosomes, the spontaneous gapping could be even more important for binding chromatin remodellers by H2A.Z containing nucleosomes.

To explore the molecular mechanism of H2A.Z-induced nucleosome gapping, we calculated the time evolution of the number of contacts of the H2A.Z N-terminal tail (residues 1–18) with DNA and examined its association with the gaping distances (Fig. 4b). We find that the notable nucleosome gapping events are coupled with the decreased number of contacts between the H2A.Z N-terminal tail and DNA. Similar transitions, showing the coupling between the H2A.Z N-terminal tail−DNA contacts and gaping motions, are observed for other simulation runs (Supplementary Fig. 15). From the conformational analysis of N-terminal tails, we see that N-terminal tails of H2A are predominantly inserted into the DNA minor and major grooves between SHL ± 3 and SHL ± 4 (Fig. 4c). These binding modes are similar to those from a previous study that performed a much more extensive sampling of histone tail conformations in canonical nucleosomes[39]. In contrast, the H2A.Z N-terminal tails have a preference to be exposed to the solvent and exhibit a relatively large conformational variability and have a reduced number of contacts with DNA (Fig. 4d). It is noteworthy that in human the H2A N-terminal tail carries a higher net positive charge (by $2e$) compared to the H2A.Z tail. Therefore, one might speculate that the H2A.Z N-terminal tail is less capable of screening DNA-DNA electrostatic repulsion between two DNA gyres as compared to the H2A tail, resulting in enhanced gapping. Interestingly, a recent study reported that the N-terminal region of H2A.Z.1 (residue 1−60) is responsible for the

flexible H2A.Z nucleosome positioning, possibly because of the reduced histone–DNA interactions[51].

## Discussion

Histone variants play crucial roles in the evolution of complex multicellular organisms and expand the function of chromatin[44,52]. It has been debated for a long time whether nucleosomes containing the H2A.Z variant are thermodynamically and dynamically stable or unstable because shreds of evidence exist in support of stabilization[53] or destabilization[25,44,54]. In the present study, we estimated the free energy barriers and performed multiple runs of seven-microsecond all-atom MD simulations to characterize the detailed atomistic mechanism by which human H2A.Z influences the dynamics and stability of nucleosomes. We find that the incorporation of the H2A.Z variant facilitates the DNA unwrapping from the octamer, enhances nucleosome gaping and the histone octamer dynamics. We also observe spontaneous DNA unwrapping of about 40 base pairs on both ends for full nucleosomes with histone tails in our all-atom MD simulations. Spontaneous DNA unwrapping in canonical nucleosomes on a time scale of tens and hundreds of milliseconds was recorded experimentally under physiological solution conditions[55–58]. The difference in time scales of these motions observed in our simulations and previous experimental studies (mostly FRET) can be explained by the higher mobility of H2A.Z nucleosomes compared to the canonical ones, by the native DNA sequence used in our study, rather than the well-positioned 601 sequence, and by overlooking the short-lived unwrapped states by FRET, biasing results to a longer lifetime scale, as reported before[56].

It has been known that tails of certain histone types are important for nucleosome stability[39,44,59–62] since truncation of those tails usually leads to the DNA unwrapping. Variant histones carry many alterations in tails including amino acid substitutions, insertions or deletions. These variations, perhaps, can attune spontaneous nucleosomal DNA breathing, unwrapping, and binding with other biomolecules. Indeed, our results show that a shorter and less positively charged C-terminal tail of human H2A.Z makes fewer interactions with the DNA ends, compared to canonical nucleosomes, and contributes to enhanced DNA dynamics and unwrapping. Simulations with the swapped H2A and H2A.Z C-terminal tails provide further evidence to demonstrate that H2A.Z tails are important for DNA unwrapping and modulating its amplitude. In addition to H2A.Z C-terminal tails, histone core plasticity is also necessary for DNA unwrapping since putting restraints on the movement of three core alpha-helices prevents some degree of unwrapping, as evident from auxiliary simulations.

Moreover, in the nucleosomal array context, the freed H2A.Z C-terminal tails may form interactions with the acidic patch from the neighboring H2A.Z nucleosome (an H2A.Z acidic patch has extra negatively charged residue(s) compared to H2A) or mediate fiber-fiber interactions as suggested previously[63]. This, in turn, can enable the formation of more compact chromatin fibers compared to the canonical systems, potentially reconciling the roles of H2A.Z in both transcription activation and facilitation of the intramolecular folding of nucleosomal arrays[64]. On the other hand, the H2A.Z N-terminal tail might be responsible for the enhanced nucleosome gaping transition explained by its decreased positive charge and therefore reduced screening of the DNA–DNA gyre repulsion.

It is reasonable to speculate that the connection between the increased mobility of H2A.Z nucleosomes and their function is to make DNA more accessible to the transcriptional machinery and other chromatin components. It has been known for some time that histone mutations (mostly involving H3 and H2A) that alter the DNA entry-exit site accessibility have direct effects on transcription elongation and termination[65]. Actually, it has been suggested previously that H2A.Z incorporation enhances terminal DNA accessibility[44,66] and therefore can facilitate transcription initiation[67,68].

Taken together, our main results provide a detailed mechanism to support the crucial role of H2A.Z in transcription following the model suggested previously[12,13,69,70]. According to the latter, the promoter-distal half of the +1 nucleosome presents a strong barrier to transcription[69,70], and the H2A.Z variant is enriched on this site and reduces the nucleosomal barrier at the transcription start site[12,13]. According to our estimates, the energy barrier is reduced by ~3 kcal/mol in H2A.Z compared to canonical nucleosomes. The major role in this mechanism is played by H2A.Z tails and by the plasticity of the nucleosome core. This enables the asymmetric unwrapping and gaping of DNA in H2A.Z containing nucleosomes. Such increased dynamics can also guide the octasome-to-hexasome transitions[71]. Overall, our study provides insights into dissecting the major structural and dynamic components in H2A.Z nucleosomes. However, the function of H2A.Z nucleosomes depends on many other factors, including the presence of histone and DNA covalent modifications and nucleosome genomic location.

## Methods

### Nucleosome modeling with the human native DNA sequence

The simulated systems in this study consist of six full nucleosome models with two straight 20-bp long DNA linker segments. The 187 bp long $(147 + 2 \times 20)$ DNA sequence is taken from *Homo sapiens TP53* gene +1 nucleosome. Note that we used this DNA sequence to construct all nucleosome models in the current study. The detailed procedure for building the nucleosome models with the native DNA sequence is described in Supplementary Materials. The X-ray structure of the human nucleosome core particle (PDB ID: 3AFA) was used to construct the canonical homotypic nucleosome containing two copies of H2A. The human homotypic H2A.Z nucleosome model was built based on the X-ray crystal structure (PDB ID: 1F66) containing human H2A.Z.1, histones H3, H4, and H2B were substituted by canonical human histone sequences. Considering that histone variant H3.3 and heterotypic histone dimers such as H2A/H2A.Z also play essential functional and structural roles[27], we also constructed structural models to include H3.3 or H2A/H2A.Z based on two other X-ray crystal structures (PDB IDs: 5B33 and 5B32). The histone tails were linearly extended into the solvent symmetrically oriented with respect to the dyad axis. The DNA sequences from the X-ray structures were substituted with the native gene sequence using the web 3DNA program[72]. As a result, we obtained four main nucleosome models (see Supplementary Fig. 1 for the initial conformations of every model): $\text{NUC}_{\text{H2A/H2A}}$ (H2A/H2A + H3/H3), $\text{NUC}_{\text{H2A.Z/H2A.Z}}$ (H2A.Z/H2A.Z + H3/H3), $\text{NUC}_{\text{H2A.Z/H2A.Z+H3.3/H3.3}}$ (H2A.Z/H2A.Z + H3.3/H3.3), and $\text{NUC}_{\text{H2A/H2A.Z+H3/H3.3}}$ (H2A/H2A.Z + H3/H3.3). For each model, three independent simulation runs were performed with each run trajectory extending up to 7 µs. To further test how the H2A/H2A.Z C-terminal tails and DNA unwrapping are correlated, we constructed extra nucleosome models by swapping the H2A C-terminal tail (residue index 120–130) of $\text{NUC}_{\text{H2A/H2A}}$ and the H2A.Z C-terminal tail (residue index 122–128) of $\text{NUC}_{\text{H2A.Z/H2A.Z}}$. Thus, two additional models are obtained (Supplementary Fig. 8): $\text{H2A}_{\text{H2AZtail}}$ (two copies of H2A with H2A.Z C-terminal tail) and $\text{H2AZ}_{\text{H2Atail}}$ (two copies of H2A.Z with H2A C-terminal tail). Note that for these two additional models, two microseconds of MD simulations are performed for each.

### Molecular dynamics simulation protocol

All MD simulations were performed with the GPU-accelerated GROMACS version 2020.6[73] using the AMBER ff14SB force field for protein and OL15 for double-stranded DNA[74,75]. Simulations were conducted in explicit solvent using an Optimal Point Charge (OPC) water model, which was recently illustrated to reproduce water liquid bulk properties and to greatly improve nucleic acid simulations as well as intrinsically disordered protein simulations[76]. The OPC water model was used in our previous long simulations of histone tails and kinetic and

thermodynamic estimates from these simulations were shown to be reasonable and agreed with observables[39]. In each simulation system, the initial structural model was solvated in a box with a minimum distance of 20 Å between the nucleosome atoms and the edges of the box. NaCl was added to the system up to a concentration of 150 mM. The solvated systems were first energy minimized using steepest descent minimization for 10,000 steps, gradually heated to 310 K throughout 800 ps using restraints, and then equilibrated for a period of 1 ns. After that, the production simulations were carried out in the isobaric-isothermic (NPT) ensemble with the temperature maintained at 310 K using the modified Berendsen thermostat (velocity-rescaling)[77] and the pressure maintained at 1 atm using the Parrinello–Rahman barostat[78]. For additional simulations with restrained histone core (referred to as H2AZ$_{restraint}$), Cα-atoms of α1, α2, and α3-helices were restrained using a harmonic potential (force constant set to 1000 kJ mol$^{-1}$ · nm$^{-2}$). A cutoff of 10 Å was applied to short-range non-bonded vdW interactions, and the Particle Mesh Ewald (PME) method[79] was used to calculate all long-range electrostatic interactions. Periodic boundary conditions were used. Covalent bonds involving hydrogens were constrained to their equilibrium lengths using the LINCS algorithm[80], allowing a 2 fs time step to be employed. For the four main simulation systems (NUC$_{H2A/H2A}$, NUC$_{H2A.Z/H2A.Z}$, NUC$_{H2A.Z/H2A.Z+H3.3/H3.3}$, and NUC$_{H2A/H2A+H3/H3.3}$), three independent simulation runs were performed (7 microseconds each) for each system and the coordinates of the solutes were collected every 100 ps yielding a total of 70,000 frames. A summary of the simulation setup is provided in Supplementary Table 1. For the three supplementary simulation systems (H2A$_{H2AZtail}$, H2AZ$_{H2Atail}$, and H2AZ$_{restraint}$), each simulation was perfromed for 2 μs and the coordinates of the solutes were collected every 100 ps yielding a total of 20,000 frames. A summary of the simulation setup is provided in Supplementary Table 2. Supplementary Table 3 shows an overview of the volume, number of particles, and simulation performance.

### Analysis of nucleosome dynamics
MD trajectory snapshots were first processed by performing a root mean square deviation fit of the C-α atoms of the histone core to the initial structure of the nucleosome. The first 200 ns frames of each simulation were excluded from the analysis. However, all-time evolution plots include the first 200 nanosecond trajectories. In-house codes were developed to quantify the DNA and histone paths, and DNA-histone interactions. The interactions were defined by two non-hydrogen atoms from DNA and histone within a distance of less than 4.5 Å. The unwrapped base pairs were defined as those in which the center of the base pair deviates more than 7 Å from the corresponding base pair in the initial structure. The radius of gyration of DNA segments was calculated using gmx_gyrate[73]. The standard free energy of unwrapping of $i$ base pairs was calculated using the relation

$$\triangle G_i^0 = -RT \ln\left(\frac{f_i}{f_{max}}\right) \tag{1}$$

where $f_i$ is the frequency of frames with $i$ unwrapped base pairs and $f_{max}$ is the maximum frequency of frames for unwrapped base pairs. Here, $f_i = N_i/N_{total}$, where $N_i$ is the number of frames with $i$ unwrapped base pairs in the MD simulation trajectory, and $N_{total}$ is the total number of frames in the trajectory. The multi-dimensional free energy landscapes were calculated using gmx_sham[73] with the radius of gyration and the number of histone-DNA contacts as collective variables. The interaction energies between the histone octamer and DNA were determined using the molecular mechanics generalized Born surface area (MM/GBSA) method[81], which is embedded in the Amber20 Package[82]. The salt concentration was assigned to 0.15 M, and the generalized Born model OBC2 was used with parameters igb = 5 and radii = mbondi2. The fast LCPO algorithm was applied to

compute an analytical approximation to the solvent accessible area of the molecule (molsurf = 0). The calculations were performed for the NUC$_{H2A/H2A}$ and NUC$_{H2A.Z/H2A.Z}$ systems based on every 1 ns frame.

The DNA locations in the nucleosome are referred to as super-helical locations (SHLs) which are defined as those sites where major grooves face the histone surface[83]. SHLs are numbered with respect to the dyad position which is set to zero (Fig. 1a).

### Reporting summary
Further information on research design is available in the Nature Portfolio Reporting Summary linked to this article.

## Data availability
The data that support this study are available from the corresponding author upon reasonable request. The source data in this study have been deposited in GitHub and are also available as a file linked with this paper. The molecular dynamics simulation trajectory data is provided in the Figshare repository at https://doi.org/10.6084/m9.figshare.21782570. Nucleosome structures used in this study include PDBID 3AFA, 1F66, 5B33, and 5B32. The hg19 human genome assembly is available from https://www.ncbi.nlm.nih.gov/assembly/GCF_000001405.13/. Source data are provided with this paper.

## Code availability
Computer code in this study is available from GitHub repository https://github.com/Panchenko-Lab/Supplementary_data_Li_et_al_2022 and Zenodo repository[84].

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

## Acknowledgements

We would like to thank Tiina Liinamaa for proofreading the paper. S.L., T.W., and A.R.P. were supported by the Department of Pathology and Molecular Medicine, Queen's University, Canada. ARP is the recipient of a Senior Canada Research Chair in Computational Biology and Biophysics and a Senior Investigator Award from the Ontario Institute of Cancer Research, Canada. ARP and SL acknowledge the support of the Natural Sciences and Engineering Research Council of Canada (NSERC) (No. RGPIN/02972-2021 ARP). This study used high-performance computational resources from Compute Canada (https://docs.computecanada.ca).

## Author contributions

S.L.: Conceptualization, methodology, software and writing—original draft. T.W.: Methodology, software, writing. A.P.: Conceptualization, writing—original draft, review & editing, supervision and funding acquisition.

## Competing interests

The authors declare no competing interests.
