## [Peer Review File · Nature Communications]

Histone variant H2A.Z modulates nucleosome dynamics to promote DNA accessibilityREVIEWER COMMENTS

Reviewer #1 (Remarks to the Author):

The paper addresses dynamics of the H2A.Z variant-containing nucleosomes and the canonical nucleosome as a control using molecular dynamics (MD) simulations, finding that the H2A.Z nucleosome shows more pronounced DNA unwrapping due to missing interaction of the H2A.Z C-terminal tail, gapping motion between two DNA gyres, and more mobility in histone octamer cores. They employed the standard and well-tested methods/force fields in the MD and performed simulations rather long and comprehensive as of this time, making the results well reliable. The negative side is that their main findings listed above do not go much beyond the well anticipated dynamics of H2A.Z nucleosomes. They do provide more structurely-detailed views of H2A.Z tails than that was anticipated, and thus are indeed valuable. But, they are not surprizing results, either.

The followings are minor/technical comments:

- 1) At the beginning of Results, authors wrote "147 bp of nucleosomal sequence" without giving more details, which are given partly in the Method and more fully in the reference cited. I think, at least, authors need to say they used two different sequences, one for the H2A.Z nuc and another for the canonical nuc.
- 2) Related to the first point, it is not clear how much difference between the MD results of the H2A.Z nuc and the canonical nuc comes from the difference in the DNA sequence and from the interaction to histones, at least logically. I guess the major difference is from the latter, but the former contribution is not ruled out, or even not analyzed.
- 3) Regarding the convergence of MD simulation, it has been discussed that the histone tail dynamics is so slow that the convergence of MD simulations is very tough. Authors need to address it.
- 4) Figures 1b and 1c (which are essentially the same contents) need error bars from some standard statistical analysis. The same comments applied to Figs. 2c, 4a.

Reviewer #2 (Remarks to the Author):

In this paper, Li, Wei, and Panchenko report on a series of molecular dynamics aimed at understanding the role of the histone variant H2A.Z on nucleosome structures and dynamics. Although the methods are standard, the length and number of replicates of each system make this a technically sound approach. Through extensive simulations, their results clearly show that H2A.Z incorporation increases DNA breathing in simulations, which is an important and interesting conclusion that helps resolves a debate to H2A.Z's role. This is an important conclusion which will be of significant interest to both the experimental and computational chromatin communities. However, when reading the paper beyond this (important) conclusion, it appeared that the other conclusions about the molecular details of H2A.Z's function are not strongly justified, and further analysis and possibly simulations are likely needed to strengthen this paper for Nature Communications. These include:

1. The authors observe a rough correlation between C-terminal tail interactions and DNA unwrapping, and use this to argue that the differences in the H2A.Z C-terminal tails are responsible for DNA unwrapping (Figure 2). While this may be the case, the possible correlations between these data (which isn't rigorously shown with statistics) do not show causation. That is, it is not convincing that it is primarily the C-terminal tails that results in DNA unwrapping. To conclusively demonstrate this conclusion, the authors would likely need to perform a series of "tail swap" simulations in which the C-terminal tails of H2A.Z and H2A are interchanged. If their conclusion is correct, then simulations of H2A.Z with the H2A tail should have DNA unwrapping similar to H2A, and simulations of H2A with

H2A.Z tails should have DNA unwrapping similar to H2A.Z.

2. The authors argue that H2A.Z incorporation increases histone core plasticity (Figures 3 and 4). While there are some increases in dynamics in the H2A.Z RMSD value, it's not clear if that is a direct effect of H2A.Z incorporation or if it is a result of DNA unwrapping and therefore a more indirect result of H2A.Z incorporation. A similar problem is noted in the data presented in Figure 4 for the discussion of gaping motions. The authors should demonstrate a causative effect between these factors and DNA unwrapping. For example, does restraining the histone core to suppress these motions reduce DNA unwrapping? Or if DNA unwrapping is suppressed through additional simulation restraints, are these motions also present?
3. The authors should perform rigorous error analysis on their data. For example, free energies of unwrapping (Figure 1c and S4) and MM/GBSA values (Table S2) should have error bars based on the three separate simulations performed per system.

More minor concerns include:

1. The paper is clearly written with a nucleosome-centric readership in mind. For example, they use the SHL terminology to describe DNA locations but never explain what this actually is. The introduction would also benefit from more details about the known structure and dynamics of the nucleosome.
2. In many cases data from the three trajectories per system are shown as what appears to be an average value with errors (such as in Figures 2a&b, 3b,c&e, and 4b). While that may be appropriate, it can hide the differences observed between the individual trajectories. The authors should include in SI the same data for each individual trajectory.
3. Figures 4b and S10 do not appear to start at 0 ns on the x-axis.
4. Table S1 should include the number of atoms in each system.
5. Detailed procedures for constructing nucleosome models should be provided in the SI and not referenced in a separate paper to aid the reader.
6. Details of the MM/GBSA calculations are sparse. What generalized born model was used? How was the surface area treated? Is the data in Table 2 all "per residue" data? Note that even if the per-residue values are added together they typically do not sum to the MM/GBSA energy of a complete unit due to differences in the GB term calculations.

Reviewer #3 (Remarks to the Author):

Li et.al presented an all-atom MD simulation study of nucleosome structure with H2A.Z and H3.3 histone variants, owing to the important role of H2A.Z histone in gene transcription. The simulations represents the first time that spontaneous DNA unwrapping for up to 40 base pairs is observed with full nucleosomes in an all-atom MD simulation. With native DNA sequence, the authors show the increased DNA unwrapping caused by the substitution of H2A for H2A.Z. In the mean time, it has been shown that the C-terminal tail of H2A.Z presents weaker interaction with DNA ends. Furthermore, increased histone octamer plasticity and nucleosome gaping are observed in H2A.Z deposited nucleosome.

It is encouraging to see that DNA unwrapping is addressed with state-of-the-art simulations at all-atom level. This work represents the best effort to understand the DNA unwrapping dynamics with computer simulation. However I have a few concerns regarding the analysis of the simulations.

1. First of all, important details are omitted regarding the analysis of MD trajectories, some of which are important to understand the presented results and conclusions. One confusing point, as an example, is that the authors claims the first 200 nanosecond of each simulation is excluded from analysis (line 330). Then a few sentences later, the total number of frames in a trajectory is specified as 60,000 (line 341), which coincides with the total simulation length (including the first 200 ns) sampled at 100 ps interval. The authors should make all technical details in preforming simulations and analysis crystal clear, as concerns such as in point #3 below may rise.

2. The author cherry-picked specific trajectory in the discussion of energy barrier associated with DNA unwrapping. As far as I can tell, the exit DNA in NUC_H2A/H2A run1 and the exit DNA in NUC_H2A.Z/H2A.Z run1 (Figure S4 a and c) are picked to form a comparison (Figure 1c) in showing the DNA unwrapping energy barrier is lowered by H2A.Z. This is wrong practice in terms of reporting simulation result. The three equivalent trajectories produced for each nucleosome variants are supposed to be treated as independent and equivalent sampling of the same ensemble. Judging from Figure 1b (assuming it's based on all three copies of the nucleosome variant, which should be clearly specified), the main conclusion would probably remain unchanged (with free energy values different from those in the text), if unwrapping free energy is evaluated over all trajectories, with both entry and exit DNA ends. This would represent a better comparison with experiments, usually seen as ensemble average.

It seems the same practice is applied in analysis of histone core plasticity and nucleosome gapping. In Figure 3b, 3e, 4b, S7, S8 and S10, it seems only one selected trajectory is presented. These observations based on individual trajectory should not be referred to as if they are common cases, without presenting statistics of all trajectories.

3. The authors claim the observation of increased histone octamer plasticity based on increased RMSD of histone core regions (Figure 3 and associated discussions). I'd like to point out that high RMSD doesn't mean more flexible structures, more dynamic. High RMSD means the structure under examination differs more from the reference, though higher variation of RMSD may indicate more dynamic structure. In the case of Figure 3b, the RMSD of H2A.Z increases from the onset until about 4000ns. After 4200ns, the RMSD is very stable, i.e. the variation of RMSD is not larger than H2A. It is plausible to say that H2A.Z is not in a low energy state at the beginning of this simulation, maybe due to the initial modeling work of nucleosome. After 4200ns, it reaches an energetically favored state, which exhibits no higher RMSD variation to canonical H2A, i.e. no higher plasticity.

4. Correlation is identified in analysing the role of H2A.Z N-terminal tail in nucleosome gapping (line 209-212). Then the authors assumed the enhanced gapping was caused by the loss of interactions between DNA and H2A.Z N-tail. I find this explanation unappealing, as even after a few hundreds of nanoseconds the N-tail-DNA contact is back to normal level, the gapping distance remains high. A more convincing theory, to me, is that the H2A.Z N-tail carries less positive charge, making it less capable of screening DNA-DNA electrostatic repulsion comparing to H2A. The less positive charge results in both loss of H2A.Z N-tail-DNA interaction, and enhanced gapping. I hope the authors can comment on this point, or provide more evidences to support their claim.

5. Are the plots in Figure 3b and 4b based on the same trajectory? If so, is there something similar in other trajectories? It is important to understand if this transition near 4000 ns is reversible. If reversible, this slow process is the reason of higher histone core plasticity. If not, the theory in point #3 is more plausible.

Additionally, the following minor points need to be addressed as well.

6. What is SHL? Though it is labeled in Figure 1a, a clear definition is lacking, which is difficult for the audiences to follow the discussions.

7. On line 100, it states "(unwrapped DNA distribution's) bimodal shape with each peak corresponding to one histone copy". How is this correspondence established? It's not clear to me.

8. The authors state the RMSD is with respect to "the equilibrated structure" on line 173, contradicting the statement on line 329 which says it's relative to "the minimized structure of the nucleosome". This should be clarified and made clear.

9. Why was the MM/GBSA calculation based on the first 200ns production trajectories (I assume it's between 200-400 ns in simulation time)? Why wasn't it based on the very last part (say 200ns) of simulations?

In conclusion, though this study presents an interesting simulation work, it is not recommended to be published in the current form, due to the flaws in the analysis, especially with respect to the histone octamer plasticity and nucleosome gapping. I would recommend this manuscript for publication if all the above concerns are addressed.

RESPONSE to REVIEWERS' COMMENTS

We sincerely thank the reviewers for their thoughtful and constructive comments and suggestions, they have improved the overall quality of our work. We have extended our previous simulations and added new ones, per reviewers' suggestions. Changes in the manuscript are highlighted in red.

Responses to reviewer #1

Overall comment: The paper addresses dynamics of the H2A.Z variant-containing nucleosomes and the canonical nucleosome as a control using molecular dynamics (MD) simulations, finding that the H2A.Z nucleosome shows more pronounced DNA unwrapping due to missing interaction of the H2A.Z C-terminal tail, gapping motion between two DNA gyres, and more mobility in histone octamer cores. They employed the standard and well-tested methods/force fields in the MD and performed simulations rather long and comprehensive as of this time, making the results well reliable. The negative side is that their main findings listed above do not go much beyond the well anticipated dynamics of H2A.Z nucleosomes. They do provide more structurally-detailed views of H2A.Z tails than that was anticipated, and thus are indeed valuable. But, they are not surprising results, either.

Response: We would like to thank the reviewer for the concise summary of our work. As we outline in the Introduction section, experimental and computational studies are very contradictory, showing that H2A.Z-containing nucleosomes can be less or more stable than the canonical ones (1-5). The detailed mechanism of the H2A.Z variant in modulating nucleosomal DNA accessibility remains unknown.

The followings are minor/technical comments:

Comment 1: At the beginning of Results, authors wrote "147 bp of nucleosomal sequence" without giving more details, which are given partly in the Method and more fully in the reference cited. I think, at least, authors need to say they used two different sequences, one for the H2A.Z nuc and another for the canonical nuc.

Response: We appreciated the reviewer's advice. We have added Supplementary Figure 2 in the revised manuscript to show the DNA sequence that was used to construct the nucleosome

models. In this study, we used the same DNA sequence (from *Homo sapiens TP53 gene*) for building all nucleosome systems including both canonical H2A (NUC_{H2A/H2A}) and H2A.Z variant (NUC_{H2A.Z/H2A.Z}) nucleosomes. We have added clarification in the revised manuscript. The detailed protocol for constructing all our nucleosome models was added in the revised Supplementary Materials.

Comment 2: Related to the first point, it is not clear how much difference between the MD results of the H2A.Z nuc and the canonical nuc comes from the difference in the DNA sequence and from the interaction to histones, at least logically. I guess the major difference is from the latter, but the former contribution is not ruled out, or even not analyzed.

Response: We should clarify that we used the same DNA sequence for constructing both canonical H2A (NUC_{H2A/H2A}) and H2A.Z variant (NUC_{H2A.Z/H2A.Z}) nucleosomes and other models (see Methods and Supplementary Materials). Therefore, we argue that the main differences between NUC_{H2A/H2A} and NUC_{H2A.Z/H2A.Z} systems, which we observed in this study, are due to the incorporation of different histones.

Comment 3: Regarding the convergence of MD simulation, it has been discussed that the histone tail dynamics is so slow that the convergence of MD simulations is very tough. Authors need to address it.

Response: We agree with the reviewer that convergence of histone tail dynamics in atomistic simulations is difficult to achieve and requires very long simulation times. Although the sampling of tail conformation was the main topic of our previous paper, to address these considerations, we performed multiple independent simulation runs for each studied system. Results from all runs point to a similar trend and the role of histone tails in DNA unwrapping and nucleosome gapping. We have made the utmost effort to utilize all our computer resources to increase the simulation time. However, performing long-time scale all-atom MD simulations on a system with ~2.7 million atoms is very time-consuming. We present a summary of our simulation systems, as well as the performance of each system using GPU-accelerated HPC clusters, in Supplementary Table 3. As one can see from this table, it takes about one year to run the simulations on several microsecond time scale on state-of-the-art computers. Nevertheless, all our main simulation runs have been extended to 7 microseconds by the time of submitting our revised manuscript. In addition, per

reviewer# 2 suggestions, we performed several additional simulations (although on a shorter timescale) to provide further evidence for the mechanism we describe and to strengthen our findings. Taken together, our total simulation time of all systems has reached 90 microseconds, which could provide sufficient evidence to support the conclusions we present in this manuscript.

Comment 4: Figures 1b and 1c (which are essentially the same contents) need error bars from some standard statistical analysis. The same comments applied to Figs. 2c, 4a.

Response: Following the reviewer's suggestions, we have added error bars, where applicable, or presented the results for all independent runs for each system in Figures 1b, 1c, 2c, and 4a.

Responses to reviewer #2

Overall comment: In this paper, Li, Wei, and Panchenko report on a series of molecular dynamics aimed at understanding the role of the histone variant H2A.Z on nucleosome structures and dynamics. Although the methods are standard, the length and number of replicates of each system make this a technically sound approach. Through extensive simulations, their results clearly show that H2A.Z incorporation increases DNA breathing in simulations, which is an important and interesting conclusion that helps resolve a debate to H2A.Z's role. This is an important conclusion which will be of significant interest to both the experimental and computational chromatin communities. However, when reading the paper beyond this (important) conclusion, it appeared that the other conclusions about the molecular details of H2A.Z's function are not strongly justified, and further analysis and possibly simulations are likely needed to strengthen this paper for Nature Communications.

Response: We would like to thank the reviewer for carefully reading the manuscript and providing the comments.

Comment 1: The authors observe a rough correlation between C-terminal tail interactions and DNA unwrapping, and use this to argue that the differences in the H2A.Z C-terminal tails are responsible for DNA unwrapping (Figure 2). While this may be the case, the possible correlations between these data (which isn't rigorously shown with statistics) do not show causation. That is,

it is not convincing that it is primarily the C-terminal tails that results in DNA unwrapping. To conclusively demonstrate this conclusion, the authors would likely need to perform a series of “tail swap” simulations in which the C-terminal tails of H2A.Z and H2A are interchanged. If their conclusion is correct, then simulations of H2A.Z with the H2A tail should have DNA unwrapping similar to H2A, and simulations of H2A with H2A.Z tails should have DNA unwrapping similar to H2A.Z.

Response: We thank the reviewer for this excellent suggestion. We performed a new set of simulations by swapping the H2A and H2A.Z C-terminal tails (Supplementary Figure 8 and Supplementary Table 2). The nucleosome systems with swapped tails are named as H2A_{H2AZtail} (two copies of H2A histones with H2A.Z C-terminal tail) and H2AZ_{H2Atail} (two copies of H2A.Z variants with H2A C-terminal tail), respectively. Due to the limited time offered for revisions and the extensive computational costs, we could not extend these simulations (~2.7 million atoms) to seven microseconds as for our main systems (which took about a year, see Supplementary Table 3 for an overview of the volume, number of particles, and simulation performance for each of our simulation system). However, we were able to simulate the dynamics of both H2A_{H2AZtail} and H2AZ_{H2Atail} for two microseconds. Our results reveal that H2A_{H2AZtail} indeed shows enhanced DNA unwrapping as compared to the NUC_{H2A/H2A} system, whereas H2AZ_{H2Atail} suppresses DNA unwrapping as compared to the NUC_{H2A.Z/H2A.Z} system (Supplementary Figure 9). These observations are in line with the conclusions we made that the C-terminal tails of H2A.Z play a crucial role in facilitating the DNA unwrapping from the histone octamer. We acknowledged the anonymous reviewer in the acknowledgment section for this very useful suggestion.

Comment 2: The authors argue that H2A.Z incorporation increases histone core plasticity (Figures 3 and 4). While there are some increases in dynamics in the H2A.Z RMSD value, it's not clear if that is a direct effect of H2A.Z incorporation or if it is a result of DNA unwrapping and therefore a more indirect result of H2A.Z incorporation. A similar problem is noted in the data presented in Figure 4 for the discussion of gaping motions. The authors should demonstrate a causative effect between these factors and DNA unwrapping. For example, does restraining the histone core to suppress these motions reduce DNA unwrapping? Or if DNA unwrapping is suppressed through additional simulation restraints, are these motions also present?

Response: The reviewer makes a very good point. To test whether the increased histone core plasticity is an effect of H2A.Z incorporation, we performed additional MD simulations with

artificially restrained histone core C α -atoms of three alpha-helices. This nucleosome system (H2AZ_{restraint}) has been simulated on a 2-microsecond time scale. Our results show that restraining histone core plasticity partially suppresses the DNA unwrapping as compared to the unrestrained NUC_{H2A.Z/H2A.Z} systems (Supplementary Figure 13, where all analyses use 2-microsecond time scale for comparison). These results provide further evidence that the increased histone core fluctuations observed in our simulations are a direct effect of H2A.Z incorporation, not DNA unwrapping. It also suggests that the enhanced DNA unwrapping of NUC_{H2A.Z/H2A.Z} happens due to both factors: H2A.Z C-terminal tails and the histone core plasticity. As for the effects of H2A.Z-induced nucleosome gapping, we find that restraining the histone core plasticity has a negligible impact on influencing nucleosome gapping motions (Supplementary Figure 17).

Comment 3: The authors should perform rigorous error analysis on their data. For example, free energies of unwrapping (Figure 1c and S4) and MM/GBSA values (Table S2) should have error bars based on the three separate simulations performed per system.

Response: We have added the error bars, where applicable, in the revised figures and MM/GBSA calculations (now Supplementary Table 4 in the revised manuscript). We have also added the results for all three simulation runs on the same figure.

Comment 4: The paper is clearly written with a nucleosome-centric readership in mind. For example, they use the SHL terminology to describe DNA locations but never explain what this actually is. The introduction would also benefit from more details about the known structure and dynamics of the nucleosome.

Response: We thank the reviewer for this suggestion. We have added the details of the definition of super-helical locations (SHL) in the Methods section of our revised manuscript.

Comment 5: In many cases data from the three trajectories per system are shown as what appears to be an average value with errors (such as in Figures 2a&b, 3b,c&e, and 4b). While that may be appropriate, it can hide the differences observed between the individual trajectories. The authors should include in SI the same data for each individual trajectory.

Response: We have added the results from three simulation runs in the Supplementary Materials.

Comment 6: Figures 4b and S10 do not appear to start at 0 ns on the x-axis.

Response: We have revised Figures 4b and S10 to start from 0 ns which includes the first 200 nanosecond trajectories. It should be noted that in our other analyses, the first 200 nanosecond frames of each simulation were excluded.

Comment 7: Table S1 should include the number of atoms in each system.

Response: Based on the reviewer's suggestion, we have added Supplementary Table 3 to include this information and other details of our simulations.

Comment 8: Detailed procedures for constructing nucleosome models should be provided in the SI and not referenced in a separate paper to aid the reader.

Response: We have added the details of the protocol for constructing nucleosome models in the Supplementary Materials in the revised manuscript.

Comment 9: Details of the MM/GBSA calculations are sparse. What generalized born model was used? How was the surface area treated? Is the data in Table 2 all "per residue" data? Note that even if the per-residue values are added together they typically do not sum to the MM/GBSA energy of a complete unit due to differences in the GB term calculations.

Response: We thank the reviewer for pointing this out. We have added the details of the MM/GBSA calculations in the Methods section of our revised manuscript.

Responses to reviewer #3

Overall comment: Li et.al presented an all-atom MD simulation study of nucleosome structure with H2A.Z and H3.3 histone variants, owing to the important role of H2A.Z histone in gene transcription. The simulations represents the first time that spontaneous DNA unwrapping for up to 40 base pairs is observed with full nucleosomes in an all-atom MD simulation. With native DNA

sequence, the authors show the increased DNA unwrapping caused by the substitution of H2A for H2A.Z. In the mean time, it has been shown that the C-terminal tail of H2A.Z presents weaker interaction with DNA ends. Furthermore, increased histone octamer plasticity and nucleosome gapping are observed in H2A.Z deposited nucleosome.

It is encouraging to see that DNA unwrapping is addressed with state-of-the-art simulations at all-atom level. This work represents the best effort to understand the DNA unwrapping dynamics with computer simulation. However I have a few concerns regarding the analysis of the simulations.

Response: We would like to thank the reviewer for these encouraging comments on our work.

Comment 1: First of all, important details are omitted regarding the analysis of MD trajectories, some of which are important to understand the presented results and conclusions. One confusing point, as an example, is that the authors claims the first 200 nanosecond of each simulation is excluded from analysis (line 330). Then a few sentences later, the total number of frames in a trajectory is specified as 60,000 (line 341), which coincides with the total simulation length (including the first 200 ns) sampled at 100 ps interval. The authors should make all technical details in performing simulations and analysis crystal clear, as concerns such as in point #3 below may rise.

Response: We thank the reviewer for pointing this out. We have corrected the simulation length for trajectory analysis. We have added the details of the protocol for constructing nucleosome models in the Supplementary Materials in the revised manuscript.

Comment 2: The author cherry-picked specific trajectory in the discussion of energy barrier associated with DNA unwrapping. As far as I can tell, the exit DNA in NUC_H2A/H2A run1 and the exit DNA in NUC_H2A.Z/H2A.Z run1 (Figure S4 a and c) are picked to form a comparison (Figure 1c) in showing the DNA unwrapping energy barrier is lowered by H2A.Z. This is wrong practice in terms of reporting simulation result. The three equivalent trajectories produced for each nucleosome variants are supposed to be treated as independent and equivalent sampling of the same ensemble.

Judging from Figure 1b (assuming it's based on all three copies of the nucleosome variant, which should be clearly specified), the main conclusion would probably remain unchanged (with free

energy values different from those in the text), if unwrapping free energy is evaluated over all trajectories, with both entry and exit DNA ends. This would represent a better comparison with experiments, usually seen as ensemble average. It seems the same practice is applied in analysis of histone core plasticity and nucleosome gapping. In Figure 3b, 3e, 4b, S7, S8 and S10, it seems only one selected trajectory is presented. These observations based on individual trajectory should not be referred to as if they are common cases, without presenting statistics of all trajectories.

Response: Per reviewers' suggestion, we have revised Figure 1c (now Figure 1e) and added error bars that present the DNA unwrapping energy barriers from both entry and exit DNA ends. We have updated the main text values accordingly. We have also revised Figures 1b, 2c and 4c to show the distributions of specific properties (unwrapped base pairs, distances) from each simulation run. For Figures 3b, 3e, 4b, S7, S8 and S10, we have added Supplementary figures showing the results from three individual trajectories in the revised manuscript. As to the time evolution plots of contacts and distances, we have tried to add all six trajectories on one figure (three for contacts and three for distances, but it compromised the visual clarity, therefore we have decided to show only one in the main figure and references others in Supplementary Materials (Supplementary figure 15 and 16).

Comment 3: The authors claim the observation of increased histone octamer plasticity based on increased RMSD of histone core regions (Figure 3 and associated discussions). I'd like to point out that high RMSD doesn't mean more flexible structures, more dynamic. High RMSD means the structure under examination differs more from the reference, though higher variation of RMSD may indicate more dynamic structure. In the case of Figure 3b, the RMSD of H2A.Z increases from the onset until about 4000ns. After 4200ns, the RMSD is very stable, i.e. the variation of RMSD is not larger than H2A. It is plausible to say that H2A.Z is not in a low energy state at the beginning of this simulation, maybe due to the initial modeling work of nucleosome. After 4200ns, it reaches an energetically favored state, which exhibits no higher RMSD variation to canonical H2A, i.e. no higher plasticity.

Response: We agree with the reviewer that high RMSD does not necessarily correspond to more flexible and dynamic structures. Therefore, we calculated the root mean square fluctuation (RMSF) of C α atoms in the globular core domain for each copy of histone (Figure 3b, 3c, Supplementary Figure 11). The results show that the H2A.Z and H2B core regions in NUC_{H2A.Z/H2A.Z} has higher

structural fluctuations compared to the canonical $\text{NUC}_{\text{H2A}/\text{H2A}}$, suggesting the overall octamer dynamics and plasticity are enhanced by the H2A.Z variant, consistent with our analysis of 2D projections of core α -helices (Supplementary Figure 12).

Comment 4: Correlation is identified in analyzing the role of H2A.Z N-terminal tail in nucleosome gapping (line 209-212). Then the authors assumed the enhanced gapping was caused by the loss of interactions between DNA and H2A.Z N-tail. I find this explanation unappealing, as even after a few hundreds of nanoseconds the N-tail-DNA contact is back to normal level, the gapping distance remains high. A more convincing theory, to me, is that the H2A.Z N-tail carries less positive charge, making it less capable of screening DNA-DNA electrostatic repulsion comparing to H2A. The less positive charge results in both loss of H2A.Z N-tail-DNA interaction, and enhanced gapping. I hope the authors can comment on this point, or provide more evidences to support their claim.

Response: We thank the reviewer for this excellent point to explain our observations, and we have added this possible explanation to the Discussion section of our revised manuscript. We have acknowledged the anonymous reviewer in the acknowledgment section for this very useful suggestion.

Comment 5: Are the plots in Figure 3b and 4b based on the same trajectory? If so, is there something similar in other trajectories? It is important to understand if this transition near 4000 ns is reversible. If reversible, this slow process is the reason of higher histone core plasticity. If not, the theory in point #3 is more plausible.

Response: Yes, the plots in Figure 3b and 4b in the original manuscript are based on the same trajectory. In the revised manuscript we have replaced Figure 3b with RMSF calculated from all three runs, per the reviewer's previous suggestion. As to Figure 4b, we do not see a reversible transition for this run on this time scale, although for two other runs (see Supplementary Materials Figure 15), we see reversible transitions around 4 microseconds.

Comment 6: What is SHL? Though it is labeled in Figure 1a, a clear definition is lacking, which is difficult for the audience to follow the discussions.

Response: We appreciated the reviewer's advice and have added the definition of super-helical locations (SHL) in the Methods section of our revised manuscript.

Comment 7: On line 100, it states "(unwrapped DNA distribution's) bimodal shape with each peak corresponding to one histone copy". How is this correspondence established? It's not clear to me.

Response: We thank the reviewer for pointing this out. We have removed this point from our revised manuscript, our initial interpretation of these peaks was not correct.

Comment 8: The authors state the RMSD is with respect to "the equilibrated structure" on line 173, contradicting the statement on line 329 which says it's relative to "the minimized structure of the nucleosome". This should be clarified and made clear.

Response: We have clarified this point in the revised manuscript.

Comment 9: Why was the MM/GBSA calculation based on the first 200ns production trajectories (I assume it's between 200-400 ns in simulation time)? Why wasn't it based on the very last part (say 200ns) of simulations?

Response: Per reviewer's suggestion, we have included MM/GBSA calculations based on the first 200 ns (200-400 ns) and last 200 ns (6800-7000 ns) of simulation trajectories (see Supplementary Table 4).

References

1. Lewis, T.S., Sokolova, V., Jung, H., Ng, H. and Tan, D.Y. (2021) Structural basis of chromatin regulation by histone variant H2A.Z. *Nucleic Acids Res.*, **49**, 11379-11391.
2. Horikoshi, N., Arimura, Y., Taguchi, H. and Kurumizaka, H. (2016) Crystal structures of heterotypic nucleosomes containing histones H2A.Z and H2A. *Open Biol.*, **6**, 160127.
3. Watanabe, S., Radman-Livaja, M., Rando Oliver, J. and Peterson Craig, L. (2013) A Histone Acetylation Switch Regulates H2A.Z Deposition by the SWR-C Remodeling Enzyme. *Science*, **340**, 195-199.

4. Weber, Christopher M., Ramachandran, S. and Henikoff, S. (2014) Nucleosomes Are Context-Specific, H2A.Z-Modulated Barriers to RNA Polymerase. *Mol. Cell*, **53**, 819-830.
5. Abbott, D.W., Ivanova, V.S., Wang, X., Bonner, W.M. and Ausió, J. (2001) Characterization of the Stability and Folding of H2A.Z Chromatin Particles: implications for transcriptional activation. *J. Biol. Chem.*, **276**, 41945-41949.

REVIEWER COMMENTS

Reviewer #1 (Remarks to the Author):

According to reviewers' comments, authors have revised the manuscript significantly. I recommend the revised manuscript to be accepted.

Reviewer #2 (Remarks to the Author):

The authors have performed extensive additional simulations and analysis which has significantly strengthened their manuscript. They have also been very responsive to the reviewers original comments. I can now recommend publication.

Reviewer #3 (Remarks to the Author):

The authors have substantially improved the manuscript by providing more rigorous analysis and new simulations. All my concerns have been addressed. I recommend this work for publication.